Evaluating the method reproducibility of deep learning models in biodiversity research

http://orcid.org/0000-0002-9354-3527 Ahmed Waqas 1 ahmed.waqas@uni-jena.de
Kommineni Vamsi Krishna 1 2 3
http://orcid.org/0000-0002-2382-9722 König-Ries Birgitta 1 2 4
http://orcid.org/0000-0002-7565-4389 Gaikwad Jitendra 1
http://orcid.org/0000-0002-8122-9522 Gadelha Luiz 1
http://orcid.org/0000-0002-7981-8504 Samuel Sheeba 1 4
1 Heinz Nixdorf Chair of Distributed Information Systems, Friedrich-Schiller Universität Jena , Jena, Thuringia , Germany
2 German Centre for Integrative Biodiversity Research (iDiv) , Leipzig, Sachsen , Germany
3 Department of Functional Biogeography, Max Planck Institute for Biogeochemistry , Jena, Thuringia , Germany
4 Michael Stifel Center Jena , Jena, Thuringia , Germany
Stowell Dan
Electronic publication date: 2025 Feb 5
Publication date: 2025
Volume: 11
Electronic Location ID: e2618
Received 2024 Jul 11; Accepted 2024 Nov 28
Copyright: © 2025 Ahmed et al.
Copyright year: 2025
Copyright holder: Ahmed et al.
License: This is an open access article distributed under the terms of the Creative Commons Attribution License, which permits unrestricted use, distribution, reproduction and adaptation in any medium and for any purpose provided that it is properly attributed. For attribution, the original author(s), title, publication source (PeerJ Computer Science) and either DOI or URL of the article must be cited.
License URL: https://creativecommons.org/licenses/by/4.0/

Keywords: Reproducibility, Deep learning, Metadata, Biodiversity

Funding: Thuringia State and the German Centre for Integrative Biodiversity Research (iDiv) Halle-Jena-Leipzig German Research Foundation (FZT 118, 202548816) This work was supported by the thurAI project funded by the Thuringia State and the German Centre for Integrative Biodiversity Research (iDiv) Halle-Jena-Leipzig, funded by the German Research Foundation (FZT 118, 202548816). The funders had no role in study design, data collection and analysis, decision to publish, or preparation of the manuscript.

==============================
Artificial intelligence (AI) is revolutionizing biodiversity research by enabling advanced data analysis, species identification, and habitats monitoring, thereby enhancing conservation efforts. Ensuring reproducibility in AI-driven biodiversity research is crucial for fostering transparency, verifying results, and promoting the credibility of ecological findings. This study investigates the reproducibility of deep learning (DL) methods within the biodiversity research. We design a methodology for evaluating the reproducibility of biodiversity-related publications that employ DL techniques across three stages. We define ten variables essential for method reproducibility, divided into four categories: resource requirements, methodological information, uncontrolled randomness, and statistical considerations. These categories subsequently serve as the basis for defining different levels of reproducibility. We manually extract the availability of these variables from a curated dataset comprising 100 publications identified using the keywords provided by biodiversity experts. Our study shows that a dataset is shared in 50% of the publications; however, a significant number of the publications lack comprehensive information on deep learning methods, including details regarding randomness.

Introduction

Biodiversity research investigates the role of biological diversity, underlying mechanisms, and means to protect it (see e.g., https://www.idiv.de/research/). This interdisciplinary research is critical for developing effective conservation strategies and guiding policy decisions that mitigate ecosystem disruptions. In recent years, deep learning methods have been increasingly applied in many scientific fields, including biodiversity research.

These methods have the potential to process and analyze large amounts of biological data rapidly, leading to significant insights. For instance, August et al. (2020) demonstrated how artificial intelligence (AI) image classifiers can create new biodiversity datasets from social media imagery, highlighting the spatial and taxonomic biases that can influence ecological inference. Similarly, deep learning (DL) models have been utilized to analyze camera trap images for wildlife monitoring, enabling researchers to identify species and infer ecological patterns and processes with high accuracy in Tabak et al. (2019). However, there is a growing concern about the reproducibility, transparency, and trustworthiness of research findings produced using deep learning methods in this domain (Feng et al., 2019; GPAI, 2022).

Reproducibility is essential in scientific research as it allows researchers to validate and advance methods and results, ensuring the reliability of scientific claims (Samuel & König-Ries, 2021). Goodman, Fanelli & Ioannidis (2016) define research reproducibility in three aspects: methods reproducibility, results reproducibility, and inferential reproducibility. In this article, we focus specifically on methods reproducibility within the deep learning analytical workflow, which involves replicating experimental and computational processes using the same data and tools to achieve consistent outcomes. While issues related to experimental design reproducibility, such as those in field studies, are important, they are outside the scope of this work. For instance, in natural studies like Rovero et al. (2013), which deploys camera traps for consistent data collection across different geographical locations, it is inherently challenging to achieve exact reproducibility due to factors such as variations in wildlife behavior or environmental conditions that affect the collected data. On the other hand, our focus remains on ensuring reproducibility within computational methods that are more controlled, such as deep learning pipelines. For example, Norouzzadeh et al. (2018) demonstrated the application of deep learning for automatically identifying and counting wildlife in camera trap images, which is critical for monitoring species populations and informing conservation strategies. However, failure to reproduce these methods accurately could lead to erroneous population estimates, thereby compromising conservation efforts and resource allocation. Furthermore, Christin, Hervet & Lecomte (2019) discuss how the reproducibility of deep learning models in ecological research is often challenged by the complexity and heterogeneity of biodiversity data, which includes interactions between variables, missing values, and non-linear patterns. Ensuring reproducibility is key to uncovering these complexities and improving the reliability of ecological models used for decision-making.

A deep learning pipeline is a structured sequence of processes in training and deploying a DL model (El-Amir & Hamdy, 2020). The pipeline typically begins with data collection and preprocessing, involving tasks such as data cleaning, normalization, and transformation. Following data preprocessing, the next stage consists of designing and selecting an appropriate deep learning architecture, considering factors like model complexity and the nature of the problem. Subsequently, model training takes place, where the chosen architecture is trained on the preprocessed data using optimization algorithms and specific hyperparameter configurations. After training, the model is evaluated and fine-tuned using various performance metrics to ensure its effectiveness in solving the targeted problem. The best-performing model is run on the test data to ensure an unbiased evaluation of the model’s predictive performance. Finally, the trained model is deployed for real-world applications or further refinement.

To ensure the reproducibility of the deep learning pipeline, comprehensive documentation is crucial at each stage. This includes detailed records of the data collection steps (including providing a persistent identifier for each data point), data preprocessing steps, such as the specific data transformation techniques applied and any data augmentation strategies employed. Additionally, it is vital to document the specifics of the chosen deep learning architecture, including the exact configurations and versions of the neural network layers utilized. Detailed notes on the hyperparameter values selected during the model training phase are essential, as well as the training convergence criteria and the optimization algorithms employed. Proper documentation of the evaluation metrics used and the testing dataset ensures the reproducibility of the model’s performance assessment. Finally, maintaining a record of the software libraries, hardware, frameworks, and versions utilized throughout the pipeline aids in replicating the experimental setup accurately.

In this article, we aim to shed light on the current state of reproducibility of deep learning methods applied to biodiversity research. We conducted a systematic literature review to identify publications that use deep learning techniques in biodiversity research using keywords provided by biodiversity experts (Abdelmageed et al., 2022). We define various reproducibility-related variables inspired by the current literature. We then curated a dataset of 100 publications and manually extracted reproducibility-related variables, such as the availability of datasets and code. We also analyzed advanced reproducibility variables, such as the specific deep learning methods employed, models, and hyperparameters. Our findings suggest that the reproducibility of deep learning methods in biodiversity research is generally low. However, there is a growing trend towards improving reproducibility, with more and more publications making their datasets and code available. Our study will contribute to the ongoing discourse on the reproducibility of deep learning methods in biodiversity research and help to improve the credibility and impact of these methods in this vital field.

In the following sections, we provide a detailed description of our study. We start with an overview of the state-of-the-art (“Related Work”). We provide the methodology of our study (“Methododology”) We describe our results and discuss the implications of our work (“Discussion”). Finally, we summarize the key aspects of our study and provide future directions of our research (“Conclusion”).

Related work

Method reproducibility, as mentioned in Goodman, Fanelli & Ioannidis (2016), is an important aspect of progress in any scientific field. Raff (2019) provides some insight into reproducibility in machine learning (ML). The author studied the reproducibility of 255 ML articles published from 1984 to 2017 and tried to correlate the reproducibility of the articles with their characteristics. According to his observation, the main obstacle to reproducibility of results is insufficient explanation of the method and implementation details when the source code is not provided. However, some of the defined attributes were very subjective, such as the algorithmic difficulty of the work or the readability of the work. While Raff (2019) provides insight into ML reproducibility in general, there is also research specifically related to the field of biodiversity (Feng et al., 2019; Schnitzer & Carson, 2016). They pointed out that reproducibility is problematic because erroneous data is widespread and there is a lack of empirical studies to verify previous research findings. Several authors have pointed out the need for better data management and reporting practices to ensure the reproducibility of research results (Stark, 2018; Waide, Brunt & Servilla, 2017; Samuel, Löffler & König-Ries, 2021). In Waide, Brunt & Servilla (2017), the authors discuss the challenges associated with managing different data sets in ecological research, while Leonelli (2018) further argues that reproducibility should be a criterion for assessing research quality. However, setting standards for assessing reproducibility itself is an important issue for research. Gundersen, Shamsaliei & Isdahl (2022) in his latest study defined 22 variables that could be categorized according to different degrees of reproducibility. Three categories were mentioned: reproducible data, reproducible experiment and reproducible method. Gundersen, Shamsaliei & Isdahl (2022) used these variables to evaluate reproducibility support on 13 different open source machine learning platforms. Similarly, Heil et al. (2021) proposed three different standards for reproducibility in machine learning that have applications in the life sciences. Bronze, silver or gold standards are defined according to the availability of data, models and code. Tatman, VanderPlas & Dane (2018) went a step further and analyzed ML articles from International Conference on Machine Learning (ICML) and NeurIPS conferences and distinguished levels of reproducibility based on the resources provided with the article. To achieve the desired result, they recommended some practical steps such as providing code and data in an executable environment where all libraries and dependencies are linked. This allows code or experiments to run smoothly on a new machine. This recommendation is in line with the gold standard provided in Heil et al. (2021) that authors should produce reproducible results with the execution of a single command. In addition to the general recommendations, Pineau et al. (2021) proposed standard practices and activities to improve reproducibility in the AI community. Some of these are the reproducibility programme at the NeurIPS conference and the ML reproducibility checklist, which provides guidelines to authors before submitting to conferences. Inspired by Pineau et al. (2021), some of the major AI conferences (AAAI and IJCAI) have introduced similar reproducibility checklists. We have considered these guidelines and also previous work in Gundersen, Shamsaliei & Isdahl (2022) to develop 10 reproducibility variables. However, the work in Gundersen, Shamsaliei & Isdahl (2022) is designed to assess the reproducibility of different ML platforms, whereas we aim to assess the reproducibility of biodiversity research. Our variables also include the aspect of uncontrolled randomness and the statistical information necessary for reproducibility.

Methodology

In this section, we will discuss the steps of our work analyzing the reproducibility of research articles in biodiversity research.

Identification

To assess the reproducibility of methods in biodiversity research, we first tried to obtain unbiased and relevant publications. To do this, we adopted keywords from Abdelmageed et al. (2022), which were provided by biodiversity experts and were also used to develop a corpus in the field of biodiversity for large language models (Abdelmageed et al., 2022). These keywords are: “biodivers*” OR “genetic diversity” OR “*omic diversity” OR “phylogenetic diversity” OR “population diversity” OR “species diversity” OR “ecosystem diversity” OR “functional diversity” OR “microbial diversity” OR “soil diversity” AND “Deep Learning”.

Using this search query, Google Scholar returned more than 8,000 articles from 2015 to 2021. Due to the impracticality of manually processing such a large dataset within the available time frame, we initially selected the first 100 articles and then randomly selected an additional 100 for further analysis. The first 100 articles were chosen with the belief that they would be statistically representative of the broader set. We compared the results of these first 100 articles with those of the randomly selected 100, which reinforced our belief, as the results were nearly identical. However, we acknowledge that this selection may not be entirely free from bias. Despite this, our sample covers a diverse range of publishers and publication years, ensuring a broad cross-section of the literature. Each selected publication was manually reviewed to extract information on ten key variables, with approximately 40 min dedicated to reviewing each article. This thorough manual curation aimed to enhance the reliability of our dataset. This manually curated dataset will serve as a benchmark for developing an automated system to extract variable information, enabling future analyses to encompass a larger number of articles while maintaining methodological rigor and reproducibility.

Screening

Before analyzing the individual articles for reproducibility, we found that some articles could not be considered further for the following reasons: 1) duplication, 2) only an abstract was provided or the full article was not accessible, and 3) some articles were not empirical studies and therefore did not include experiments. After considering all these limitations, the number of articles was reduced to 100. Figure 1 shows the distribution of articles over the period from 2015 to 2021, with most articles being from 2020 and 2021. Figure 2 shows the publisher information for the considered articles, which are distributed quite randomly across more than ten publishers.

Figure 1 Number of publications analyzed for the presence of reproducibility variables, categorized by year.

Figure 2 Publisher information for the 100 publications selected for reproducibility analysis.

Selection of reproducibility variables

There is no standard for assessing the reproducibility of DL biodiversity methods. A common practice for analyzing the reproducibility of research articles is to rerun the experiments using the same methodology as the original author. However, this requires a lot of time and computing resources. Also, sometimes it is not possible to work with the same resources such as hardware and software. Inspired by Gundersen, Shamsaliei & Isdahl (2022), we have formulated a set of 10 indicators that can be used as proxies for the probability that a result is reproducible. Instead of repeating each experiment, we look for these variables or factors that are considered important determinants of the reproducibility. Using the literature and the reproducibility checklists at conferences such as NeurIPS and AAAI (Pineau et al., 2021), we can divide these variables into four categories. Resource information (ReI) details the availability of datasets, source code, open-source tools, and proprietary model details crucial for reproduction efforts. Methodological information (MI) captures the specifics of software and hardware used, as well as a high-level overview of the deep learning methods employed. Randomness Information (RaI) addresses aspects of unpredictability in computational processes, ensuring they are documented for consistency. Statistical information (SI) focuses on the rigor of result analysis, advocating for multiple metrics and averaging techniques for reliable evaluation. The comprehensive details that define these aspects of reproducibility are systematically itemized in Table 1.

Table 1 Definition of the various reproducibility variables used in this research article.

Category	Variable	Variable short name	Variable description	
Resource Information (ReI)	Dataset	V1	Availability of dataset with persistent identifiers.	
	Source code	V2	Availability of source code to recreate the experiment (e.g., GitHub, GitLab, Zenodo).	
	Open source frameworks or environment	V3	Availability of open source software and hardware tools required to reproduce the work (e.g., docker container, virtual environments).	
	Model architecture	V4	Availability of a deep learning model architecture or the accessibility of its internal working.	
Methodological Information (MI)	Software and Hardware Specification	V5	Availability of information related to the type of hardware used (e.g., GPU), its specifications, and the version of software and libraries used.	
	Methods	V6	Availability of high-level information about the deep learning pipeline including the pre-processing and post-processing steps. The other parameters of the pipeline are assessed as separate variables.	
	Hyper-parameters	V7	Availability of hyperparameters used to train the model (e.g., number of epochs, learning rate, optimizer, etc.).	
Randomness Information (RaI)	Randomness	R	Availability of information about weight initialization, data shuffling, data augmentation, data train-test split.	
Statistical Information (SI)	Averaging result	S1	Availability of multiple model training and averaging them out rather than selecting only the highest value.	
	Evaluation metrics	S2	Availability of more than one evaluated metric (e.g., R2 score along with Root Mean Square Error (RMSE) or Mean Absolute Error (MAE) or Loss of the model).	

Assessing the reproducibility of published articles

For all selected articles, we first manually checked the variables for each article. To reduce the degree of subjectivity, we (two of the authors) began by independently assessing the availability of each reproducibility variable. In the initial phase of our assessment, we encountered notable discrepancies due to ambiguous interpretations of the definitions of each variable, leading to inconsistent results from both authors. This was quantitatively evidenced by an average Cohen’s Kappa value (Cohen, 1960) of 0.57, indicating a moderate level of agreement and highlighting the initial inconsistencies between annotators. To address these issues and improve the reliability of our analysis, we undertook a review and clarification of the variable definitions. In the end, we obtained the same responses for each article, as shown in Table A1. These responses formed the basis for analysing reproducibility. We verified whether the article covered the functionality of each previously defined variable. After completing the assessment for each article, we quantified reproducibility across five distinct levels. These levels are designed to provide independent researchers with insight into the likelihood of successfully replicating the results. The higher the level of reproducibility, the more variables are accounted for, increasing the chances of reproducing the results with accuracy. The levels are structured to reflect the time and computational effort needed for reproducibility, especially when certain variables are missing. However, meeting the criteria for the basic level (Level 1) is essential for a study to be considered reproducible, as defined by Goodman, Fanelli & Ioannidis (2016). Levels beyond this build upon Level 1, offering increasingly rigorous criteria, but Level 1 remains the foundational requirement. Figure 3 illustrates the different levels of reproducibility and the corresponding categories each covers.

Level 1 covers all variables defined in the ReI category, and a study is classified as potentially reproducible once this level is met.

Level 2 includes all variables from both the ReI and MI categories, classifying a study as mostly reproducible when these criteria are fulfilled.

Level 3 expands further, covering ReI, MI, and statistical information (SI).

Level 4 incorporates ReI, MI, and randomness information (RaI). These two levels do not necessarily build upon each other, offering additional insights without being required for basic reproducibility.

Level 5, the highest level, encompasses all categories, ensuring the most comprehensive reproducibility criteria.

Figure 3 Number of levels along with the categories covered for respective levels.

This tiered structure ensures flexibility and allows for varying degrees of reproducibility to be recognized. While Level 1 is the minimum threshold, Levels 2 and above provide additional layers of robustness, offering greater confidence in the study’s replicability. We believe this approach balances thoroughness with practicality, while also accounting for studies that may be classified as mostly reproducible without needing to meet every criterion at the higher levels.

Results

As discussed in “Discussion”, we divided the ten reproducibility variables we had defined into four categories: 1) resource information 2) methodological information 3) randomness information, and 4) statistical information. These four categories were further categorized into five different levels of reproducibility, allowing us to assess the reproducibility level of the publications more comprehensively.

Variable level information

Figure 4 contains four plots, one for each category of information. Within each category, specific reproducibility variables are depicted with binary counts, illustrated by two side-by-side bars: one denoted by ‘Yes’ in a brick orange color and the other by ‘No’ in blue, along with the corresponding number of publications. In total, we made 1,000 individual judgements for the ten variables, 579 of these were positive i.e the respective information was provided, while this was not the case in 421 cases resulting in negative judgements. Detailed descriptions of all responses can be found at both the category and individual variable levels.

Figure 4 Presence of the considered reproducible variables in four categories: (A) resources, (B) methodological information, (C) randomness, (D) statistical consideration for selected research publication.

In the resources information category, 50 publications provided a dataset, 30 indicated the code repositories, 88 used open-source frameworks or environments, and 96 mentioned the model architectures (Fig. 4A).

For the methodological information category, around a quarter of the publications included details about the hardware and the software (libraries) they used, all the publications explained methods that were used to build a machine learning pipeline and 77 publications provided the basic hyperparameters (Fig. 4B).

The number of publications that used a random seed in all possible ways (weight initialization, data shuffling, data augmentation, data train-test split and cuDNN GPU library) in their code is 3 (Fig. 4C). Regarding statistical considerations, 71 publications provided the average result of multiple model trainings, 85 publications evaluated their models with more than one evaluation metric. (Fig. 4D).

Categorical level information

We have harmonized the presence of reproducibility variables for each categorical level in such a way that all the individual reproducibility variables in a category must be available (Yes) to mark that the specific categorical information is available (Yes). If one of the individual reproducibility variables is unavailable (No), the specific categorical information is unavailable (No). Mathematically, V1&V2&V3&V4==Yes is used to denote the availability of resource information (Yes).

Table 2 shows the number of publications where all the variable information is available in a publication within the defined categories. The number of publications that satisfy the Resources, Methodological, Randomness, and Statistical information categories are 22, 20, 3, and 26, respectively.

Table 2 Number of selected research publications that contain all the variables information in a specific category along four categories.

Category	Publication count	
Resource information	22	
Methodological information	20	
Randomness information	3	
Statistical information	26	

Reproducibility levels of publications

As per the definitions of the reproducibility levels described in “Methodology”, Levels 1 and 5 are the lowest and highest, respectively. According to Fig. 5, Only one publication meets the highest level of reproducibility, while 22 publications fulfill the basic criteria for being classified as “potentially reproducible.” Additionally, 12 publications meet the Level 2 criteria, covering both ReI and MI categories, and are classified as “mostly reproducible”.

Figure 5 Bar plot indicating the number of publications satisfying the different levels of reproducibility.

Reproducibility status of publications by year

The number of publications that meet the defined reproducibility criteria follows a linear trend with respect to year (Fig. 6).

Figure 6 Number of articles meeting criteria for the four categories: (A) resources, (B) methodological information, (C) randomness, (D) statistical consideration for selected research publications by year.

Discussion

In biodiversity research, deep learning methods are becoming part of many studies that run large-scale experiments. These gave us the opportunity to orchestrate and extract the 10 reproducibility variables from 100 publications (Table 1 and Table A1). As a result, we recorded 1,000 total responses: 579 were positive, and 421 were negative. All the positive responses were dominated by four variables (230 responses): 1) Open source frameworks or environment, 2) Model architecture, 3) Methods, and 4) Evaluation metrics and the negative responses were dominated by only one variable: Randomness.

Most of the publications that employed deep learning models use open-source frameworks or environments like Tensorflow and PyTorch with the programming language Python/R, and they also provide model architectures either as a figure/table in the publication or described in the text with respective citations. Some of the publications used licensed programming languages like Matlab, comparatively, it is negligible. We looked for high-level information on the whole deep-learning pipeline in methods, all the publications provided compact information.

Most of the publications use more than one metric to evaluate their models, for example, when it comes to regression tasks, they use R2 score along with root mean square error (RMSE) or mean absolute error (MAE) or loss of the model, etc. The information from the publications is in compliance with reproducible workflow guidelines for the positively dominated variables.

We found datasets only in 50 publications; in the other 50 publications, there was no tangential information about the dataset. There were some cases where the authors linked to some data-providing websites to find the data that was used in the publication, but those websites will change over time and finding the exact data that was used in the publication is not possible without providing persistent identifiers of the respective data points.

Source code is one of the fundamental variables of reproducibility. However, 30% of the publications provided their source code.

In 21 publications, authors have provided specifications about both software and hardware. Without specific information about the hardware and software, the reproducibility results will change because the random generators work differently with different hardware and software changes with each version.

Hyperparameters are the values that are chosen to control the model learning process. This means with each set of hyperparameters, the model will provide a varied range of results. However, information about basic hyperparameters (epochs, learning rate, optimizer and loss function) was missing from 23 publications. Reproducing the results of these 21 publications is difficult due to the missing essential hyperparameter information.

Averaging results is also an important aspect of reproducibility, after each training process, results will change slightly because of the certain random initializations through the deep learning pipeline. However, in our study, 71 publications did not report the multiple training results.

We opted for manually updating the variables by going through each publication and extracting the required information for the presence of reproducibility variables because we did not find a system or technique that could automatically extract this information from a publication. Since we are extracting the information about variables from publications manually, it is only possible to work with a small dataset, which is also the limitation of this study.

Due to recent developments with Large Language Models (LLMs), we are considering extracting the reproducible variables information from publications using LLMs from the year 2022 onwards (Ahmed et al., 2023; Kommineni, König-Ries & Samuel, 2024). This will allow us to implement our analysis on large-scale publications.

Conclusion

In this article, we presented our pipeline for assessing the reproducibility of deep learning methods in biodiversity research. Inspired by the current state of the art, we established a comprehensive set of ten variables, categorized into four distinct groups, to effectively quantify the reproducibility of DL empirical research. Based on the defined categories, we documented the availability of each variable across 100 selected publications over the period from 2015 to 2021. From the total 1,000 responses for the 10 variables, 57.9% show the availability of the variables in the publication, while the remaining 42.1% are primarily characterized by the absence of randomness-related information. The highest and lowest reproducibility levels are satisfied by only one and ten publications, respectively. Given the use of deep learning to advance biodiversity research, improving reproducibility of the DL methods is crucial. Considering the limitations of the manual approach and the relatively small dataset analyzed until 2021, our future endeavors will focus on implementing a semi-automatic approach that leverages Large Language models for extracting information on reproducible variables from publications.

Appendices

Table A1 presence of different reproducibility variables. ‘y’ denotes the presence of variable information, while ‘n’ signifies the absence of variable information.

Publication	V1	V2	V3	V4	V5	V6	V7	R	S1	S2	
Klein, McKown & Tershy, 2015	n	n	y	n	n	y	n	n	n	n	
Khalighifar et al. (2021)	n	n	y	y	n	y	n	n	n	y	
Choe, Chi & Thorne, 2021	n	n	y	y	y	y	y	n	y	y	
Mahmood et al. (2016)	y	n	y	y	n	y	n	n	n	y	
Younis et al. (2020)	y	y	y	y	y	y	y	n	y	y	
Schwartz & Alfaro (2021)	y	y	y	y	y	y	y	n	y	y	
Chen et al. (2020)	n	n	y	y	n	y	n	n	n	y	
Fujisawa et al. (2023)	y	y	y	y	n	y	y	n	y	y	
Weinstein (2018)	n	y	y	y	y	y	y	n	n	y	
Chalmers et al. (2021)	n	n	y	y	y	y	n	n	n	y	
Guirado et al. (2019)	y	n	y	y	y	y	y	n	y	y	
Zualkernan et al. (2020)	n	n	y	y	n	y	y	n	n	y	
Villon et al. (2018)	n	n	y	y	n	y	y	n	y	y	
Fairbrass et al. (2019)	y	y	y	y	y	y	y	n	n	y	
Weinstein et al. (2019)	y	y	y	y	y	y	y	n	n	y	
Hu et al. (2020)	n	n	y	y	n	y	n	n	n	y	
Alshahrani et al. (2021)	y	n	y	y	n	y	y	n	y	y	
Villon et al. (2020)	n	n	y	y	n	y	n	n	y	y	
Botella et al. (2018)	y	n	y	y	n	y	y	n	y	y	
Salamon et al. (2017)	y	n	y	y	n	y	y	n	y	y	
Mac Aodha et al. (2018)	y	y	y	y	y	y	y	n	y	y	
Schindler & Steinhage (2021)	n	n	y	y	n	y	y	n	n	y	
Bjerge et al. (2021)	n	y	y	y	n	y	y	y	y	y	
Guirado et al. (2017)	y	y	y	y	y	y	y	n	n	y	
Hussein et al. (2021a)	n	n	y	y	n	y	y	n	n	y	
Rammer & Seidl (2019)	y	y	y	y	y	y	y	n	n	y	
Anand et al. (2021)	n	n	n	y	n	y	n	n	n	y	
Zizka et al. (2021)	y	y	y	y	n	y	y	y	n	n	
Demertzis, Iliadis & Anezakis, 2018	n	n	y	y	n	y	n	n	y	y	
Malerba, Wright & Macreadie, 2021	y	n	y	y	n	y	y	n	n	n	
Huang & Basanta (2021)	n	n	y	y	n	y	y	n	y	y	
Campos-Taberner et al. (2020)	y	n	n	y	n	y	n	n	n	y	
Rousset et al. (2021)	y	n	y	y	n	y	y	n	n	y	
López-Jiménez et al. (2019)	y	n	y	y	n	y	y	n	n	y	
Schuettpelz et al. (2017)	n	y	y	y	n	y	y	n	n	y	
Schiller et al. (2021)	y	y	y	y	y	y	y	y	y	y	
Heredia (2017)	n	y	y	y	n	y	y	n	y	y	
Martins et al. (2021)	y	n	y	y	y	y	y	n	n	y	
Browning et al. (2018)	y	n	y	y	n	y	n	n	y	y	
Guirado et al. (2020)	y	n	y	y	n	y	y	n	n	y	
Ayhan et al. (2020)	n	n	y	y	n	y	y	n	n	y	
Loddo, Loddo & Di Ruberto, 2021	n	n	y	y	n	y	n	n	y	y	
Jamil, Rahman & Haider, 2021	y	n	y	y	n	y	y	n	n	y	
Neves et al. (2021)	n	n	y	y	n	y	n	n	n	y	
Jin et al. (2021)	y	n	y	n	n	y	n	n	n	y	
Mohanty, Hughes & Salathé, 2016	y	y	y	y	n	y	y	n	n	y	
Xie et al. (2019)	y	n	y	y	n	y	y	n	y	y	
Tian et al. (2020)	n	n	n	n	n	y	n	n	y	y	
Gimenez et al. (2022)	n	y	y	y	y	y	y	n	n	y	
Ortega Adarme et al. (2020)	y	n	y	y	n	y	y	n	n	y	
Dunker et al. (2021)	n	n	y	y	n	y	n	n	n	y	
Villon et al. (2016)	n	n	y	y	n	y	n	n	n	y	
Dyrmann et al. (2021)	n	n	y	y	n	y	y	n	n	y	
Arruda et al. (2021)	y	n	y	n	n	y	y	n	n	n	
Kislov & Korznikov (2020)	y	n	y	y	n	y	y	n	n	y	
Hussein et al. (2021b)	y	n	y	y	n	y	y	n	n	y	
Becker et al. (2023)	n	y	y	y	y	y	y	n	n	y	
Padubidri et al. (2021)	y	y	y	y	n	y	y	n	n	y	
Carranza-Rojas et al. (2017)	y	n	y	y	n	y	y	n	n	y	
Hamdi, Brandmeier & Straub, 2019	n	n	y	y	n	y	y	n	n	y	
Ali et al. (2021)	n	n	y	y	n	y	y	n	y	y	
Norouzzadeh et al. (2020)	y	y	n	y	n	y	y	n	n	y	
Sun et al. (2019)	n	n	y	y	n	y	y	n	n	y	
Carranza-Rojas, Mata-Montero & Goeau (2018)	n	n	y	y	n	y	y	n	n	y	
Torney et al. (2019)	y	y	y	y	n	y	y	n	y	y	
Rowe et al. (2021)	n	n	y	y	n	y	y	n	y	y	
Cota et al. (2021)	y	y	y	y	n	y	y	n	n	y	
Hussain et al. (2021)	y	n	y	y	n	y	y	n	n	n	
Alkhelaiwi et al. (2021)	n	n	y	y	n	y	y	n	n	y	
Norouzzadeh et al. (2018)	n	y	y	y	n	y	y	n	n	y	
Zhou et al. (2021)	y	n	y	y	y	y	y	n	y	n	
Karar et al. (2021)	y	n	y	y	n	y	y	n	n	y	
Ferreira et al. (2020)	y	y	y	y	y	y	y	n	n	y	
Monedero et al. (2021)	n	n	y	y	n	y	y	n	n	n	
Gerovichev et al. (2021)	y	y	y	y	n	y	y	n	n	y	
Pu, Xv & Deng, 2021	n	n	y	y	n	y	y	n	n	y	
Tang, Zhang & Zhao, 2021	n	n	y	y	n	y	y	n	y	y	
Morgan & Braasch (2021)	n	n	y	y	n	y	y	n	y	y	
Milosević et al. (2020)	n	n	y	y	n	y	n	n	n	n	
August et al. (2020)	y	y	y	y	n	y	n	n	n	n	
Wagner et al. (2020)	y	y	y	y	y	y	y	n	n	y	
Hass & Jokar Arsanjani (2020)	y	y	y	y	y	y	y	n	n	y	
DeLancey et al. (2019)	y	n	y	y	n	y	y	n	n	y	
Spiesman et al. (2021)	n	n	n	y	n	y	y	n	n	y	
Jiang et al. (2023)	y	y	y	y	y	y	y	n	n	y	
Charles et al. (2021)	n	n	y	y	n	y	y	n	n	y	
Pouliot et al. (2021)	n	n	y	y	n	y	y	n	n	y	
Marre et al. (2020)	n	n	y	y	y	y	y	n	n	y	
Radig et al. (2021)	n	n	n	y	n	y	n	n	n	y	
Yubo et al. (2020)	y	n	n	y	n	y	y	n	y	y	
Liang et al. (2020)	n	n	n	y	n	y	n	n	n	n	
Banzi & Abayo (2021)	y	n	y	y	n	y	y	n	n	n	
Yang et al. (2021)	n	n	n	y	n	y	y	n	n	y	
Villon et al. (2021)	n	n	n	y	n	y	n	n	n	n	
Malik, Faisal & Hussein, 2021	y	n	n	y	n	y	y	n	n	y	
Khalighifar et al. (2019)	y	y	y	y	n	y	n	n	y	n	
Loulidi et al. (2022)	n	n	n	y	n	y	y	n	n	y	
Xi et al. (2019)	n	n	y	y	n	y	y	n	y	n	
Kankane & Kang (2021)	n	n	y	y	n	y	y	n	n	y	
Lee et al. (2017)	y	y	y	y	n	y	y	n	n	n	

Supplemental Information

Supplemental Information 1 Dataset and Source code Repository.

Supplemental Information 2 DOI (Zenodo).

Additional Information and Declarations

Competing Interests

Luiz Gadelha is an Academic Editor for PeerJ Computer Science.

Author Contributions

Waqas Ahmed conceived and designed the experiments, performed the experiments, analyzed the data, performed the computation work, prepared figures and/or tables, authored or reviewed drafts of the article, and approved the final draft.

Vamsi Krishna Kommineni conceived and designed the experiments, performed the experiments, analyzed the data, performed the computation work, prepared figures and/or tables, authored or reviewed drafts of the article, and approved the final draft.

Birgitta König-Ries conceived and designed the experiments, authored or reviewed drafts of the article, and approved the final draft.

Jitendra Gaikwad conceived and designed the experiments, authored or reviewed drafts of the article, and approved the final draft.

Luiz Gadelha conceived and designed the experiments, authored or reviewed drafts of the article, and approved the final draft.

Sheeba Samuel conceived and designed the experiments, analyzed the data, authored or reviewed drafts of the article, and approved the final draft.

Data Availability

The following information was supplied regarding data availability:

The code and data are available at GitHub and Zenodo:

- https://github.com/fusion-jena/Reproduce-DLmethods-Biodiv

- Kommineni, V. K., Ahmed, W., König-Ries, B., Gaikwad, J., Gadelha, L., & Samuel, S. (2024). Evaluating the method reproducibility of deep learning models in the biodiversity research [Data set]. Zenodo. https://doi.org/10.5281/zenodo.13987177.

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
