# Peer review of "Evaluating the method reproducibility of deep learning models in biodiversity research"

_PeerJ Computer Science, doi:10.7717/peerj-cs.2618_

## Round 0.1 · original submission · Minor Revisions

The reviewers agree that this paper is interesting, well-motivated and potentially very useful contribution. There are various items of feedback from both reviewers, all of which are useful, and so I expect all comments to be addressed in a resubmission.

Reviewer 1 ·

Basic reporting

Overall the manuscript is of a high standard, with good use of English and a suitable structure. There were places I struggled to understand the methods, and this could do with some additional clarification.

1. The methods section needs greater clarity, particularly in the section ‘Reproducibility check’. I initially thought the subsection title referred to the test of your own labelling reproducibility, but perhaps it refers to the broader method for checking the other papers? I would recommend a section preceding this that clearly outlines what data is being extracted from each paper, as otherwise it is not clear what the ‘binary response’ is. I also find the description of the levels confusing.

I think one issue here, which also occurs in the discussion, is the use of the definite article with ‘binary response’, which gives the impression that each article contains an element called ‘binary response’ that the authors are searching for. For instance at line 246 “These gave us the opportunity to orchestrate and extract the binary responses of 10 reproducibility variables from 61 publications” would be clearer as something like “we assessed the presence of 10 reproducibility variables…”.


2. L41-60 I find this framing a little confusing, as it talks about experimental design as a factor in reproducibility. Yet this seems to be outwith the scope of the paper that focuses on reproducibility within the deep-learning analytical workflow. I think this tight focus is a good thing, but it would be better to clearly define the scope of the paper early. For instance there is no consideration of the inherent problems with reproducibility when conducting natural studies (for instance in the study cited in this paragraph, you’d be very unlikely to get the same camera trap images and therefore the exact same results even if you applied the exact same experimental processes perfectly), so it just needs to be made clear what is being considered here, and a little rephrasing of this paragraph could greatly improve clarity.

3. I personally find the term ‘biodiversity domain’ jarring (and note it becomes ‘Biodiversity field’ by the Discussion), as I tend to think of biodiversity as a subject of study (e.g. see discussion of definitions here https://cfcul.mcmlxxvi.net/GI/FCV/seminarios/EXSY_ReadingGroup/docs/DeLong.pdf) rather than the name of a research field - for instance ecology, biology, conservation science. I would therefore recommend the authors consider using alternative terminology, but this may just be personal preference.

4. I find Figures 4-6 unclear. I think the use of grouped instead of stacked barplots would improve interpretability and remove the need for the ‘overlap’ category, unless I’m missing something. In Figure 5 I cannot understand why the percentages in the pie chart add up to 100 but the fractions do not add up to 1 - there must be a better way to represent this data.

Experimental design

This study is within the scope of interest for the journal, addressing a clear research question and using rigorous research methods for the most part. I do have some concern about the methods used to obtain a sample of the literature.

1. It is surprising that this paper is based on a relatively small sample size, 100 papers, as the literature review returned over 8000 results. Given the small sample, it is important that it is as representative as possible. It is therefore highly surprising, and potentially problematic that the authors chose to use the top 100 results returned by Google Scholar, stating that they ‘believe’ these would be no different to the following results. Yet search algorithms select relevance on a wide number of factors that might cause substantial bias in those highly ranked (e.g. https://www.mdpi.com/1999-5903/13/2/31, https://link.springer.com/article/10.1007/s43681-022-00136-w). It would be better if the author’s could provide evidence for their belief that their sample is not biased, either from the literature or by analysing a subset of journals outside the top 100 and making a statistical comparison.

The screening process reduced the dataset down to just 61 journals or 0.7% of the original 8000 search results. There is no justification given as to why 100 was initially chosen (beyond time-restrictions for manual effort) or whether it is a suitable sample size to obtain the power required for the analysis. It is unclear how this further reduction in sample size would reduce the study power, or why journals eliminated during the screening process couldn’t be replaced by those lower in the search rankings that met the suitability criteria. Furthermore, of the 61 studies, at least 8 came from preprint servers. Whilst in a comprehensive analysis it might be suitable to include some studies that have not been peer-reviewed, having 13.5% of the sample from preprints does seem to further limit the inferences you can draw about peer-reviewed science.

Validity of the findings

The underlying data has been provided, and if the sample is shown to be represntative, then the data support the results. I do have some doubts over the clarity of the figures reporting the results, which are set out in section 2.

Additional comments

I enjoyed reading this paper, which I think addresses an important knowledge gap and provides some very interesting insights. However, I do have some concerns both about the sampling method, and about the clarity of methods and results that should be addressed prior to publication.

·

Basic reporting

Strengths:

- Language is clear throughout, article is well structured, self-contained, and contains clear definitions of all terms when needed.

- Related work is well-cited, and gives a thorough overview of past reproducibility studies that motivate the methodology pursued by the authors.

Weaknesses:

- The “Overlap” category in Figure 4 is confusing. I think it would be more clear to separate each variable into two side-by-side bars for Yes/No. The current format makes it appear as if “Overlap” is a third class.

Minor comments:
- Small typo on L69 - should be one sentence I believe

Experimental design

Strengths:

- To the best of my knowledge, this appears to meet the standards of the journal and represents original primary research.

- The research question — how reproducible are biodiversity-related deep learning publications — is well-defined and meaningful. The authors describe relevant work investigating reproducibility in both ecology and machine learning, but note that there is a gap at the intersection. Experimental design for evaluating reproducibility is well-informed by prior work.

- Investigation is rigorous, e.g.: “we manually reviewed each selected publication  to extract relevant information on ten defined variables, dedicating approximately 40 minutes per paper” (L152-L153), “To reduce the degree of  subjectivity, we (two of the authors) began by independently recording the binary responses” (L184-L185).

- In particular the taxonomy of reproducibility variables proposed is thorough, logical, and can serve as both a rigorous evaluation of existing work as well as clear guideline for future reproducibility.

- Methods are described in sufficient detail.

Weaknesses:

- The Randomness In- formation (RaI) category seems too stringent, particularly the requirement for cuDNN GPU library. It would be more appropriate to consider reporting the CUDA version as part of the V5 category. The cuDNN version will be implied by the combination of GPU hardware and CUDA version, and is not usually specified in publications, thus I don’t believe this should count against reproducibility evaluation as it can be inferred. It is also not clear how the evaluation of randomness criteria was performed. Weight initialization is an implementation detail that — while important — is often carried out in a default manner by the underlying model framework being used, and could be inferred from specifying the proper software versions as in V3/V4. Further, it is not specified how this evaluation was conducted: was the source code referenced, or only the details specified in publications? Setting random seeds is standard practice that again may be performed behind the scenes by the deep learning framework being used and may not need to be reported for reproducibility.

- The requirement in S2 to report more than one metric does not seem appropriate for evaluating reproducibility. There may only be one appropriate metric for the task at hand. It is also not clear what was considered an “evaluation metric” here — would mean + standard deviation over several random trials (as in S1) count as one metric or two?

Validity of the findings

Strengths:
- Overall, the conclusions are well-stated and supported by the experiments conducted.

- All underlying data and methods have been provided.

Weaknesses:

- There is some concern that the 61 papers surveyed are not representative of the entire intersection of biodiversity and deep learning, particularly because both fields contain a multitude of possible keywords that could have been missed by the Google Scholar query described in L142-L145. In particular, deep learning papers are more likely to describe the particular task performed by the model (e.g., image classification, object detection, object counting) rather than use the phrase ‘deep learning’ directly. Including additional keywords could lead to a more unbiased sample of deep learning based papers in biodiversity. I do believe the findings still represent a meaningful sample of the literature, and the authors also note the limitations transparently in L156.

- As noted above, the experimental design is perhaps overly stringent for evaluating reproducibility, so the conclusions are quite pessimistic in their evaluation of the current levels of reproducibility in the field. There would be room for a “mostly reproducible” category, or something similar, to indicate that many but not all of the requirements were met.

Additional comments

Overall I commend the authors for this thorough investigation. I hope the authors will consider the appropriateness of some of the reproducibility criteria proposed and perhaps relax the requirements where noted to accommodate reasonable levels of reproducibility given current standard practices in the open source deep learning ecosystem. Recognizing there is no established standard and there is a level of subjectivity here, I believe the criteria are mostly justified except where noted. I recommend accepting the manuscript with minor revisions.

---

## Round 0.2 · Minor Revisions

The reviewer is happy with your updated paper, as am I. Note that in standard English we do not use the definite article for a phrase like

"...in the biodiversity research" - it should be
"...in biodiversity research".

Thus please remove the word "the" from the end of the TITLE as I have indicated, as well as updating the first paragraph as the reviewer indicates.

·

Basic reporting

Compared to the initial versions, I find the updated figures much more clear. My concerns in this regard have been addressed. I also that addressing R1's concerns have led to more clarity, however there is still some minor inconsistency between "biodiversity research" and "biodiversity field" remaining (L279).

I find the first paragraph of the manuscript to be a bit strange now for two reasons: (1) The use of the definite article with "The biodiversity research", and the non sequitur sentence about the presence of the preprint.

Experimental design

Thank you for addressing the concerns regarding CUDnn versions and randomness information. I believe the updates to these criteria improve the manuscript.

While I still somewhat disagree with the reasoning behind S2, I think I find it acceptable in the context of S1, which would by nature imply two metrics (mean and standard deviation) would be reported to illustrate the level of randomness in the results.

Validity of the findings

The clarifications on the "Levels" are very helpful.

In regards to the concerns about the validity of findings from this relatively small sample of papers, also brought up by R1, I think the methodology is overall sound, especially considering the newly added papers did not change conclusions significantly.

---

## Round 0.3 · accepted · Accept

Thank you for your attention to detail, and resolving the remaining review comments.